# How Would The Viewer Feel?
# Estimating Wellbeing From Video Scenarios

**Mantas Mazeika**[*]
UIUC

**Eric Tang**[*]
UC Berkeley

**Andy Zou**
UC Berkeley

**Steven Basart**
UChicago

**Jun Shern Chan**
UC Berkeley

**Dawn Song**
UC Berkeley

**David Forsyth**
UIUC

**Jacob Steinhardt**
UC Berkeley

**Dan Hendrycks**
UC Berkeley

## Abstract

In recent years, deep neural networks have demonstrated increasingly strong abilities to recognize objects and activities in videos. However, as video understanding becomes widely used in real-world applications, a key consideration is developing human-centric systems that understand not only the content of the video but also how it would affect the wellbeing and emotional state of viewers. To facilitate research in this setting, we introduce two large-scale datasets with over 60,000 videos manually annotated for emotional response and subjective wellbeing. The Video Cognitive Empathy (VCE) dataset contains annotations for distributions of fine-grained emotional responses, allowing models to gain a detailed understanding of affective states. The Video to Valence (V2V) dataset contains annotations of relative pleasantness between videos, which enables predicting a continuous spectrum of wellbeing. In experiments, we show how video models that are primarily trained to recognize actions and find contours of objects can be repurposed to understand human preferences and the emotional content of videos. Although there is room for improvement, predicting wellbeing and emotional response is on the horizon for state-of-the-art models. We hope our datasets can help foster further advances at the intersection of commonsense video understanding and human preference learning.

## 1   Introduction

Videos are a rich source of data that depict vast quantities of information about humans and the world. As deep learning has progressed, models have begun to reliably exhibit various aspects of video understanding, including action recognition (Kay et al., 2017a), object tracking (Zhao et al., 2021), segmentation (Huang et al., 2019; He et al., 2020), and more. However, vision models do not exist in a vacuum and will eventually require social perception abilities, so models need to begin understanding how humans interpret and respond to visual inputs. As video models become more widely used in real-world applications, they should be able to reliably predict not only "what is where" in a visual input but also predict how it would make a human feel.

The subjective experience of human viewers on video data is broadly valuable to characterize and predict. When humans pursue goals in the world, their actions are often driven by intuitive processes (Kahneman, 2011), a significant part of which is the experience of emotions or affective states (Oatley et al., 2006). Emotions can be thought of as evaluations of events in relation to goals (Scherer et al., 2001; Frijda, 1988), and hence are important to study in relation to behavior in diverse settings. However, they are also important to understand in their own right, as they are strong indicators of what people value (Hume, 1739). For example, if a situation makes one feel happy, then that is often preferred to a situation that induces feelings of fear. Additionally, emodiversity—the variety

---

[*]Equal Contribution.

36th Conference on Neural Information Processing Systems (NeurIPS 2022) Track on Datasets and Benchmarks.

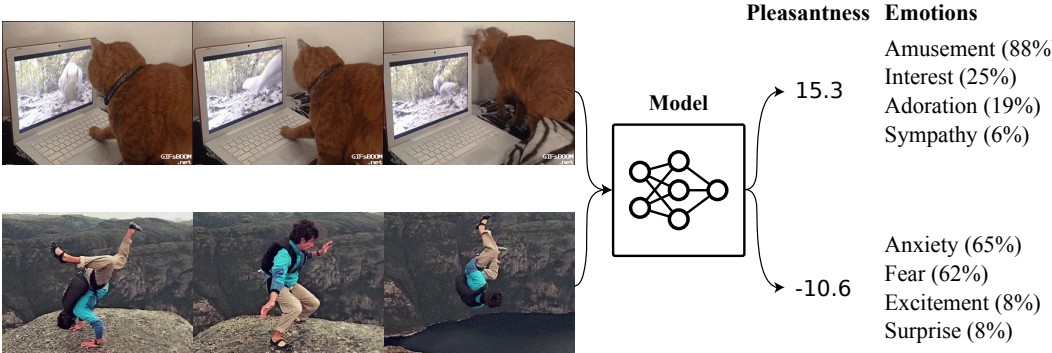

Figure 1: We introduce two large-scale datasets for predicting subjective responses to videos, including relative pleasantness between videos and distributions of fine-grained emotional responses. This enables training state-of-the-art vision models to predict continuous, consistent scores for the pleasantness of videos and a rich distribution of likely emotional responses.

of experienced emotions—is an important indicator of the overall health of the human emotional ecosystem, and emodiversity over positive emotions can improve mental wellbeing (Quoidbach et al., 2014). Thus, understanding the emotional responses and preferences of humans on video data could be a useful avenue toward modeling basic human desires, values, and overall wellbeing.

Video recommender systems already attempt to capture human preferences over videos but for practical reasons often base their recommendations on imperfect proxy metrics (Ridgway, 1956). It is hard to directly measure the values of users and how video content affects their wellbeing. Thus, recommender systems often rely on metrics that are easier to obtain, such as engagement and watch time. This simplifies the problem but can result in unintended consequences and safety concerns (Hendrycks et al., 2021b). Simplifying metrics loses sight of the experiencer (Scott, 1999) and can result in situations where engagement is maximized but users are unhappy (Russell, 2019; Kross et al., 2013; Facebook; Stray, 2020; Stray et al., 2021). For instance, content that evokes feelings of envy or anger can be highly engaging but is nonetheless unhealthy to be constantly exposed to. Thus, systems that recommend videos could substantially improve user experience through content-based inferences about how it would affect the emotional state and wellbeing of viewers.

To facilitate research on understanding how viewers feel while watching videos, we introduce two large-scale datasets for predicting emotional state and wellbeing of viewers directly from videos. First, we introduce the Video Cognitive Empathy (VCE) dataset for predicting fine-grained emotional responses to videos. The VCE dataset contains approximately 60,000 videos with human annotations for 27 emotion categories, ranging from the six basics (joy, sadness, fear, disgust, anger, surprise) (Ekman, 1992) to more nuanced emotions such as admiration and awkwardness, altogether covering the spectrum of affective states (Cowen and Keltner, 2017). As emotional responses can be considered evaluations of events in relation to a person's unique goals, they can vary significantly across human viewers. To capture the diversity of human responses, we collect a distribution—not just a single label—of emotional responses for each video. This enables evaluating models on their ability to inclusively predict the likely range of responses to a video across our large pool of annotators.

To estimate how videos affect the wellbeing of human viewers, we introduce a second dataset, Video to Valence (V2V). The V2V dataset contains approximately 25,000 videos with human-annotated rankings of pleasantness between videos. Pleasantness captures the overall positive or negative affect that viewers feel when watching a video and serves as a measure of wellbeing (Sidgwick, 1907; de Lazari-Radek and Singer, 2017). Since our annotations are for pairwise or listwise comparisons across videos, we can train utility-style models to predict continuous wellbeing scores (Hendrycks et al., 2021a), capturing gradations of wellbeing rather than a binary indicator. For instance, two scary videos may both be unpleasant, but our dataset enables predicting which video is more unpleasant, enabling a deeper understanding of human preferences.

Our datasets come with strong baselines. We train state-of-the-art video Transformers (Vaswani et al., 2017) on our tasks and find that these models, which are primarily used for understanding the literal content of videos, can predict the subjective state of viewers with surprising reliability. Although there is room for improvement, models that understand how viewers feel when watching

| Dataset | Annotation Type | Number of Videos |
|---|---|---|
| COGNIMUSE (Zlatintsi et al., 2017) | affective labels | 7 |
| HUMAINE (Douglas-Cowie et al., 2007) | affective labels | 50 |
| FilmStim (Schaefer et al., 2010) | affective labels | 70 |
| DEAP (Koelstra et al., 2012) | affective labels, face video | 120 |
| VideoEmotion (Jiang et al., 2014) | discrete emotions | 1,101 |
| LIRIS-ACCEDE (Baveye et al., 2015) | valence, arousal | 160 |
| EEV (Sun et al., 2020) | performative expressions | 5,153 |
| Video Cognitive Empathy (Ours) | fine-grained emotions | 61,046 |
| Video to Valence (Ours) | relative pleasantness | 26,670 |

Table 1: Comparisons between datasets for predicting the subjective states that human viewers would feel while watching videos. We introduce two new datasets with substantially more scenarios than prior work. Our datasets are annotated with subjective self-reports, enabling high-quality evaluations.

videos are on the horizon and may thus prove useful in numerous applications. Our datasets and experiment code can be found at github.com/hendrycks/emodiversity. We hope our datasets can help foster further research into the important problem of understanding human emotions and wellbeing.

## 2 Related Work

**Video Understanding With DNNs.** Much work in video understanding has focused on identifying various aspects of the scenarios depicted in videos. These include recognizing human motion and actions (Schuldt et al., 2004; Kuehne et al., 2011; Soomro et al., 2012; Wang et al., 2014; Karpathy et al., 2014; Caba Heilbron et al., 2015; Abu-El-Haija et al., 2016; Kay et al., 2017b; Goyal et al., 2017; Zhang et al., 2019), arbitrary event recognition (Monfort et al., 2019), spatial localization and tracking (Yilmaz et al., 2006; Milan et al., 2016; Kang and Wildes, 2016; Vondrick et al., 2018), and video segmentation (Pont-Tuset et al., 2017; Xu et al., 2018; Garcia-Garcia et al., 2018). Some work focuses on recognizing emotions and goals expressed by humans in videos, including facial emotion recognition (Lyons et al., 1998; Lucey et al., 2010; Bargal et al., 2016; Li and Deng, 2020) and recognizing unintended actions (Epstein et al., 2020). Numerous video models have been proposed and benchmarked on tasks for understanding "what is where" in videos (Gorelick et al., 2007; Tran et al., 2015, 2018; Feichtenhofer et al., 2019; Sharir et al., 2021). However, relatively little work has investigated the context in which videos are often consumed—namely, that humans watch videos and have subjective experiences deriving from said videos. Our work focuses on this important, less explored area of study.

**Predicting Subjective Responses.** Predicting the subjective responses of humans to various stimuli is an important topic of study spanning numerous fields. The International Affective Picture System (IAPS) (Lang and Bradley, 2007) and Open Affective Standardized Image Set (OASIS) (Kurdi et al., 2017) both contain approximately 1,000 images selected to evoke a range of emotional responses. Achlioptas et al. (2021) explore affective explanations of paintings as a source of training for deep learning. Eliciting emotions in text is harder, although many works have investigated predicting emotions expressed by writing (Strapparava and Mihalcea, 2007; Oberländer and Klinger, 2018; Demszky et al., 2020). Unlike still images and text, video is better suited to studying subjective responses, as video stimuli can be far more evocative. Numerous datasets have been proposed to study emotional responses to video (Zlatintsi et al., 2017; Douglas-Cowie et al., 2007; Schaefer et al., 2010; Koelstra et al., 2012; Jiang et al., 2014; Baveye et al., 2015; Sun et al., 2020). Notably, Cowen and Keltner (2017) collect self-reported emotional states on a bank of 2,185 online videos and find that reported emotional states factor into 27 distinct emotions, which we use as a framework for building our VCE dataset, which is $30\times$ larger. Comparisons of our datasets to existing work are given in Table 1. Our datasets have a much greater scale and diversity of videos than prior work, enabling research on predicting subjective responses with state-of-the-art deep learning models.

**Value Learning.** Building machine learning systems that interact with humans and pursue human values may require understanding aspects of human subjective experience. Many argue that values are derived from subjective experience (Hume, 1739; Sidgwick, 1907; de Lazari-Radek and Singer, 2017) and that some of the main components of subjective experience are emotions and valence. Learning

| Video Cognitive Empathy (VCE) Examples | Top Emotions |
|---|---|

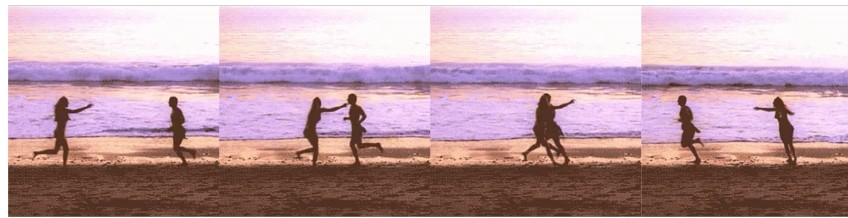

Amusement (79%)
Surprise (21%)
Awkwardness (14%)
Romance (14%)
Empathic Pain (7%)

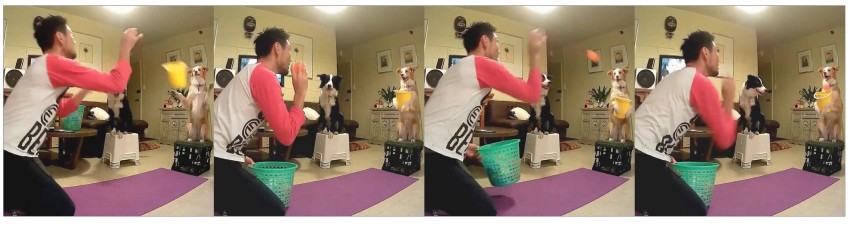

Amusement (77%)
Admiration (23%)
Adoration (23%)
Interest (23%)
Surprise (8%)

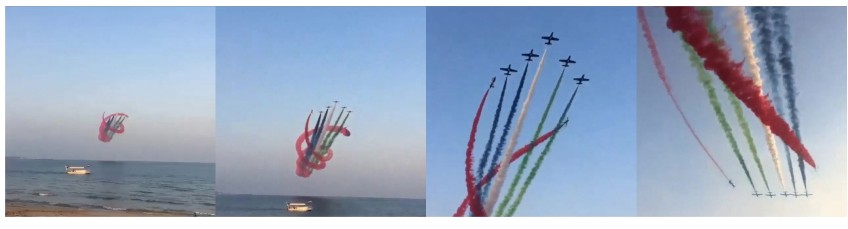

Awe (36%)
Entrancement (36%)
Aesthetic Appr. (27%)
Surprise (27%)
Interest (23%)
Satisfaction (18%)

Figure 2: Examples from the Video Cognitive Empathy (VCE) dataset. Each video is annotated with a distribution of emotional responses from forced choice decisions across multiple annotators. We ask whether models can predict emotional responses solely from the semantic content of videos.

representations of values is necessary for creating safe machine learning systems (Hendrycks et al., 2021b) that operate in an open world. In natural language processing, models are trained to assign wellbeing or pleasantness scores to arbitrary text scenarios (Hendrycks et al., 2021a). Recent work in machine ethics (Anderson and Anderson, 2011) has translated this knowledge into action by using wellbeing scores to steer agents in diverse environments (Hendrycks et al., 2021c). However, this recent line of work so far exclusively considers text inputs rather than raw visual inputs.

**Emodiversity.** A large body of work in psychology seeks to understand and quantify the richness and complexity of human emotional life (Barrett, 2009; Lindquist and Barrett, 2008; Carstensen et al., 2000). An important concept in this area is emodiversity, the variety and relative abundance of emotions experienced by an individual, which has been linked with reduced levels of anxiety and depression (Quoidbach et al., 2014). Although prior work studies emodiversity in self-reports of emotion without stimuli, we hypothesize that the emodiversity of visual stimuli may be an important concept to quantify and understand. Thus, we investigate how our new datasets enable measuring the emodiversity of in-the-wild videos on a large scale.

## 3   Video Cognitive Empathy (VCE) Dataset

When watching videos, humans feel a wide range of emotions based on the semantic content depicted in the video. These emotional responses may depend on the video in complex ways, requiring reasoning about the implications of depicted events as well as a robust understanding of human values. We are interested in whether deep models can exhibit cognitive empathy, the ability to understand how someone else is feeling or would feel in a certain situation. To test whether state-of-the-art video models can predict emotional responses, we introduce the Video Cognitive Empathy (VCE) dataset.

**Dataset Description.** The VCE dataset contains 61,046 videos with annotations for the emotional response of human viewers. The data are split into a training and test set of 50,000 and 11,046 videos, respectively. Each video lasts an average of 14.1 seconds for a total of 239 hours of manually

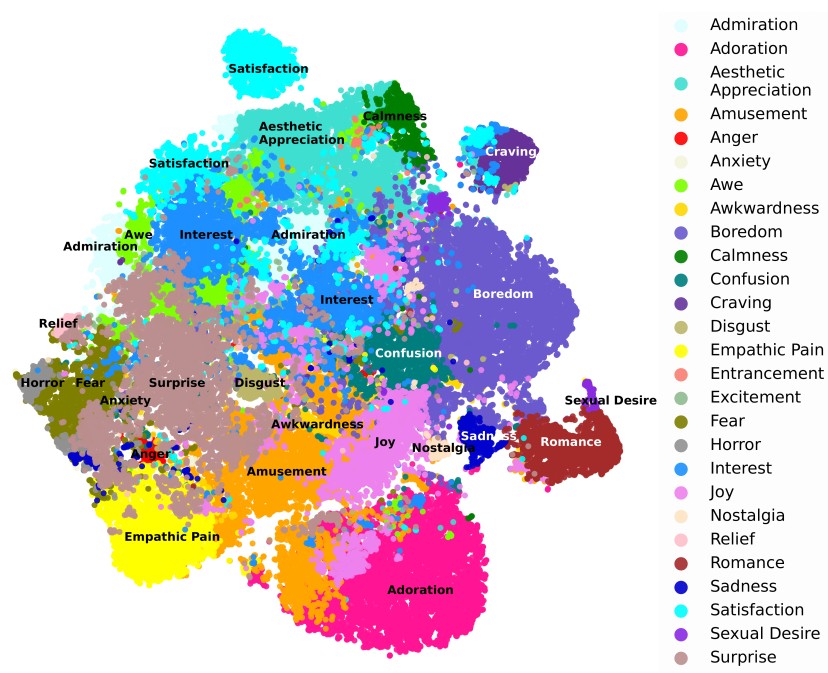

Figure 3: t-SNE plot of all 27-dimensional annotation vectors in the Video Cognitive Empathy dataset. Points are colored according to the most prevalent evoked emotion. Groups of emotions cluster together in natural ways, allowing for intuitively reasonable traversals through the space of emotions.

annotated data. While movies often evoke emotions with soundtracks and appropriate choices of colors and lighting, we are interested in how emotions depend on the semantic content of videos and less so on how engineered cues can evoke desired emotions. Thus, we remove audio cues that could serve as confounding variables. We also filter out inappropriate videos using an automated nudity detector followed by manual filtering for each video based on video thumbnails. VCE is the first dataset of its size with manual annotations that is suitable for evaluating modern deep video models.

The annotations in VCE are modeled after the analysis performed by Cowen and Keltner (2017). By collecting reported emotional experiences from humans on a set of 2,000 videos, they find that emotional responses exhibit 27 dimensions associated with reliably distinct situations. These correspond to 27 descriptive emotional states, such as "admiration", "anger", and "amusement". We adopt this fine-grained categorization of emotions and ask annotators to indicate which emotions they felt the most while watching a video. In Figure 2 of the Supplementary Material, we show the number of annotations per emotion.

As emotional responses can vary across annotators, we capture the distribution of responses by gathering a large number of annotations per video. For each video in VCE, we gather an average of 13 annotations (minimum of 12, maximum of 15). Rather than only keeping examples with high inter-annotator agreement, which would result in a small dataset, we consider the distribution of responses to be the target for learning. This is justified because while individual emotional responses are variable, the distribution of emotional responses tends to change with the stimuli. For example, scary movies might not scare everyone, but the dominant response is fear. However, responses to certain content such as political videos can vary considerably across populations. Hence, our annotations should not be taken to be representative of all emotional responses and are primarily intended for studying whether deep networks can acquire cognitive empathy.

**Dataset Construction.** Annotations for VCE were collected using Amazon Mechanical Turk (MTurk) with IRB approval. For each video, workers were asked to view the video without audio and select from the set of 27 emotions the emotions that the video most strongly evoked. For each selected emotion, workers were asked to rank the intensity of that emotion from 1 to 10. To ensure that labels are high quality, we required that MTurkers pass a qualification test, and provided them with detailed

definitions of each of the 27 emotions. We also ensured that workers viewed the entire video, only worked on one task at a time, and asked workers to mark videos that would rely too heavily on audio in order to rate.

Note that the MTurk annotators and individuals depicted in the videos may not form a representative sample of diverse cultural backgrounds. Hence, our annotations should not be taken to represent accurate emotional responses across a broad range of cultures or on an individual level, and we discourage their use in deployment contexts. The VCE and V2V datasets are designed to give a high-level understanding of how well current video models can predict subjective responses to videos. We support work on large-scale data collection that considers differences in emotional responses across cultures and individuals, and we think this is an interesting direction for future research.

### 3.1 Metrics

We evaluate models on VCE using a top-$k$ accuracy metric. Let $(x, y) \in \mathcal{D}$ be a sample video and annotation. The annotation $y$ is a 27-dimensional vector with non-negative entries that indicates the intensity of responses for each of the 27 emotion categories. Let $f(x)$ be the predicted output distribution of a model $f$ on video $x$. The top-$k$ accuracy is computed as $\frac{1}{|\mathcal{D}|} \sum_{(x,y) \in \mathcal{D}} \mathbb{1} \left[ \arg \max f(x) \in [\arg \operatorname{sort} y]_{-k:} \right]$, where argsort is in ascending order and the colon notation indicates the last $k$ indices of the resulting array. This measures the fraction of test examples where the maximum predicted emotion is in the top $k$ emotions of the ground-truth distribution. We set $k = 3$ for our evaluations.

### 3.2 Analysis

**Emotion Clusters.** Cowen and Keltner (2017) find that emotions vary continuously and cluster in reasonable ways. For example, one can smoothly traverse their 27-dimensional space of emotions by going from calmness to aesthetic appreciation to awe. To investigate whether our responses exhibit this behavior, we perform dimensionality reduction on the 27-dimensional VCE response distribution using t-SNE. We visualize results in Figure 3. Points are colored according to the maximum emotion in the response distribution. We find that emotions cluster together and that clusters group in natural ways. The groupings exhibit smooth transitions similar to Cowen and Keltner (2017). For example, one can smoothly transition through calmness $\rightarrow$ aesthetic appreciation $\rightarrow$ awe, and adoration $\rightarrow$ amusement $\rightarrow$ surprise. This demonstrates that the distributions of emotional responses contain significant hidden information beyond the top emotion for a given video.

**Emodiversity.** The emotion distribution labels in VCE enable measuring per-video emodiversity. Emodiversity is an indicator of the overall health of the human emotional ecosystem and is positively correlated with mental wellbeing (Quoidbach et al., 2014). Prior work measures emodiversity using the Shannon entropy (Quoidbach et al., 2014), but this metric can be hard to interpret (Magurran, 2003). Thus, we quantify emodiversity using perplexity of the normalized emotion distribution, computed as $\exp\left(-1 \cdot \sum_i y_i \log y_i\right)$ where $y_i$ is the normalized probability assigned to the $i^{\text{th}}$ emotion in video label $y$. One may also exclude negative emotions like disgust when computing emodiversity, since emodiversity over positive emotions is more relevant to im-

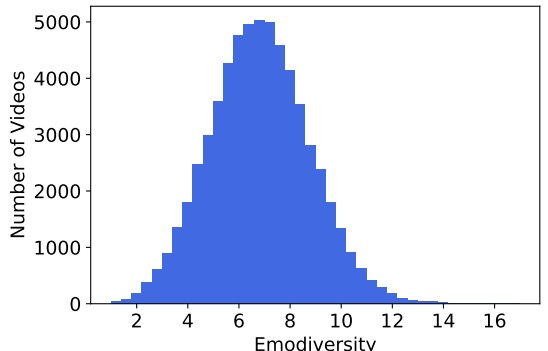

Figure 4: The emodiversity of videos in VCE. The value of the emodiversity can be interpreted as the number of emotions that a video evokes. Most videos evoke a wide range of emotions across viewers.

proving wellbeing. This metric has a minimum of 1 and a maximum of the number of emotion clusters. It can be interpreted as the number of emotions that a video evokes, assuming uniform responses for all evoked emotions. In Figure 4, we show the distribution of emodiversity across VCE. This shows that most videos evoke a diverse range of emotions across the population of viewers.

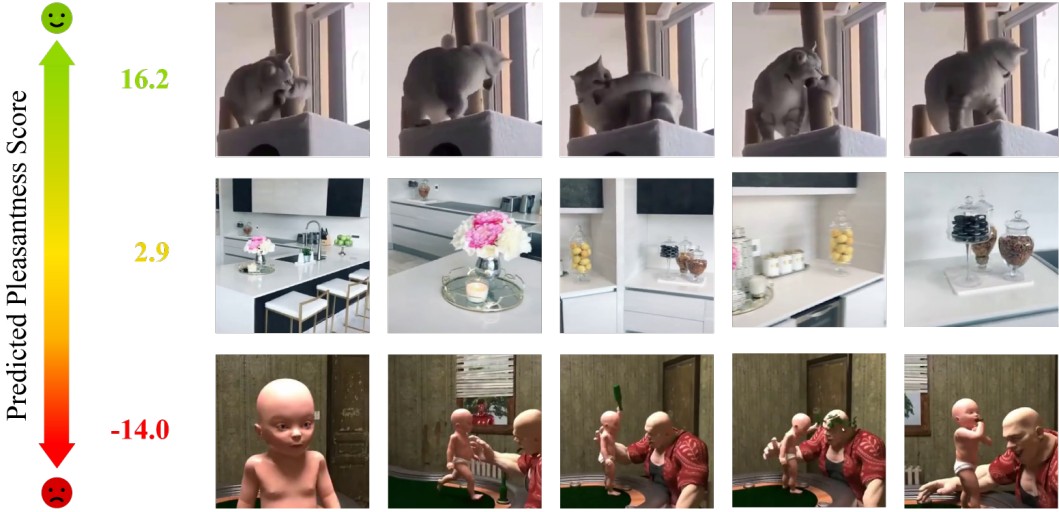

Figure 5: Example predictions on the V2V dataset. We train video models to predict continuous pleasantness scores by enforcing consistency with pleasantness rankings in the V2V training set. This results in intuitively reasonable outputs that capture preferences over the content depicted in videos.

## 4 Video to Valence (V2V) Dataset

A defining attribute of many emotional states is valence, which indicates how positive or negative an emotion is. For instance, feelings of joy typically have high valence, and feelings of fear typically have low valence. In addition to cognitive empathy via fine-grained prediction of which emotions are likely to be felt on a video, we also want video models to have a robust understanding of how a video would affect the valence of viewers' emotional state and by extension their overall wellbeing.

An important and underexplored characteristic of valence is that it varies continuously. Even within emotions such as fear, some experiences can be more pleasant or preferable than others. Thus, simply binning videos as "positive" or "negative" is a vast oversimplification that misses substantial portions of human experience. To enable developing robust models of gradations of wellbeing experienced while watching videos, we introduce the Video to Valence (V2V) dataset.

Video to Valence (V2V) Pleasantness Ranking Example

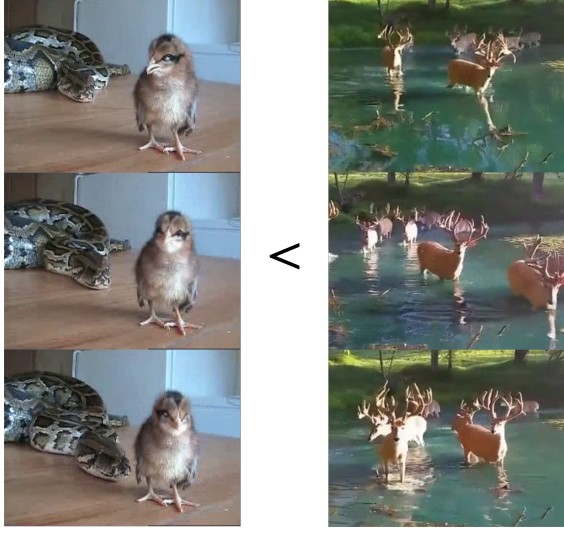

Figure 6: An example video pair in the Video to Valence (V2V) dataset. The annotators have high agreement that the video on the left is less pleasant than the video on the right.

**Dataset Description.** The V2V dataset contains 26,670 videos with annotations for rankings of pleasantness across videos. The data are split into a training and test set of 16,125 and 10,545 videos, respectively. The training set contains 11,038 pairwise annotations, and the test set contains 4,947 pairwise and listwise annotations. Each video lasts an average of 14.3 seconds for a total of 106 hours of manually annotated data. As in VCE, we are interested in how subjective state depends on the semantic content of videos rather than on audio or lighting cues. Additionally, the videos in V2V are a subset of VCE, enabling a richer analysis of the interplay between fine-grained emotional states and rankings of pleasantness.

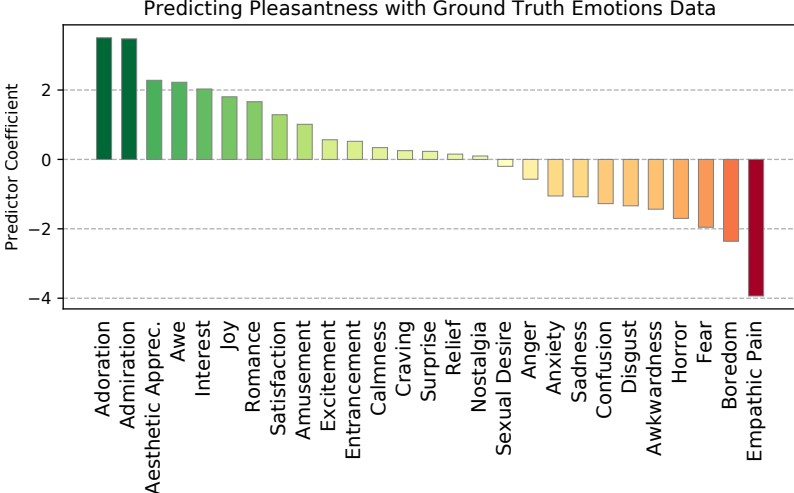

Figure 7: The coefficients from a linear model that predicts video valence (V2V) from emotions data (VCE). The emotions that contribute most strongly to pleasantness have higher positive coefficients and vice versa. This provides evidence that predicting emotional responses and estimating wellbeing are complimentary tasks that can benefit from being studied together.

The annotations in V2V are for relative pleasantness between videos. Compared to binary pleasantness, relative pleasantness enables building models of gradations of wellbeing that capture much more detail about what people value. Additionally, rankings on pairs of videos are more repeatable and consistent across annotators than alternatives such as Likert scales. Accordingly, we find that annotators have much higher agreement rates for ranking the pleasantness of videos than for reporting fine-grained emotional responses. Consequently, all the annotations in V2V are for clear-cut comparisons with a high agreement rate across 9 independent annotations.

When annotating relative pleasantness between pairs of videos, an important consideration is ensuring that comparisons are informative and interesting. For example, comparing videos that primarily evoke joy and videos that primarily evoke fear introduces very little novel information, as joy is preferable to fear for most people. In natural language datasets, one can simply construct counterfactual scenarios where slight differences have large effects on valence (Hendrycks et al., 2021a). However, this strategy is not currently viable for videos. Thus, we choose a balanced sampling strategy that selects pairs of videos based on multiple criteria, including similarity between emotional responses. Consequently, the construction of V2V depends on the VCE annotations. Additional details are in the Supplementary Materials.

**Dataset Construction.** Annotations for V2V were collected using MTurk with IRB approval. We required workers to pass a qualification test and monitored agreement rate among workers over time, dropping workers who appeared to be selecting more randomly. We collected 9 pairwise annotations for each video pair, keeping annotations that 8 or 9 distinct workers agreed on. We first collected 6 pairwise annotations for each pair, then paused labeling for pairs that already had high disagreement. For the remaining high agreement pairs, 3 more labels were collected, after which the pair was either added to the dataset or discarded.

### 4.1 Metrics

We evaluate models on V2V using the accuracy of predicted pairwise comparisons. Let $(i, j) \in \mathcal{I}$ be a set of indices in our dataset with a pairwise comparison, where video $i$ is less pleasant than video $j$ by convention. Let $x_i, x_j \in \mathcal{X}$ be corresponding videos, and let $y_{ij} \in \mathcal{Y}$ be the pairwise label, where $y_{ij} = 0$ if video $i$ is more pleasant than video $j$ and $y_{ij} = 1$ if video $j$ is more pleasant than video $i$. Let $f(x_i, x_j)$ be the prediction of model $f$ for the pairwise label. Pairwise accuracy is computed as $\frac{1}{|\mathcal{I}|} \sum_{(i,j) \in \mathcal{I}} \mathbb{1}\left[f(x_i, x_j) = y_{ij}\right]$.

As V2V has a substantial number of pairwise comparisons, it is possible to consider the pairwise comparisons between one video and multiple other videos. Thus, we also evaluate models on

their ability to correctly predict the most pleasant video in lists of $n$ videos with overlapping annotations. Let $(i_1, i_2), (i_2, i_3), \ldots, (i_{n-1}, i_n) \in \mathcal{I}$ be a list of overlapping annotations. Let $\mathcal{I}^*$ be the set of all such listwise comparisons, possibly with different values of $n$. Listwise accuracy is computed as $\frac{1}{|\mathcal{I}^*|} \sum_{L \in \mathcal{I}^*} \prod_{(i,j) \in L} \mathbb{1}\left[f(x_i, x_j) = 0\right]$, which corresponds to the fraction of lists on which the model correctly identifies the ground-truth preference ordering for the entire list. We use $n \in \{3, 4\}$. Listwise accuracy is a more challenging metric than pairwise accuracy and evaluates how well the model simultaneously predicts relative pleasantness across larger ranges of the input space.

| Method | Performance |
|---|---|
| STAM | 66.4% |
| VideoMAE | 68.9% |
| R(2+1)D | 65.6% |
| Majority Emotion | 35.7% |
| CLIP | 28.4% |

Table 2: Emotion prediction results on VCE. All models outperform random chance (11.1%), and Video Transformers have the highest accuracy.

## 4.2 Analysis

Since V2V videos are a subset of VCE videos, we can analyze how the two tasks are related. A particularly interesting question is whether binary pleasantness is sufficient to predict ranking annotations in V2V. We do not directly collect binary pleasantness annotations, so we operationalize positive valence as the value of the "joy" emotion in VCE annotations. We train a logistic regression model using this unidimensional feature and find that performance on the V2V test set is near chance, at 51% pairwise accuracy. This indicates that the mere presence of positive emotions is insufficient for predicting gradations of valence.

To analyze the importance of the full distribution of emotional responses, we repeat the above experiment with all 27 emotions as features. In this case, pairwise accuracy increases to 89.6%, indicating that the information encoded by multiple emotions can be combined to predict pleasantness rankings with high accuracy. To analyze the behavior of this model, we plot the logistic regression weights for each emotion in Figure 7. The learned weights make intuitive sense; high-valence emotions have large weights, and low-valence emotions have low weights. This suggests that distributions of emotional responses can serve as strong features for predicting continuous measures of wellbeing.

## 5 Experiments.

**Models.** *STAM* (Sharir et al., 2021) samples a small number of input frames throughout the video and aggregates across time with global attention; we use STAM-16 by default. *VideoMAE* (Tong et al., 2022) adapts masked autoencoder pretraining of vision Transformers (He et al., 2022) to the video domain. *R(2+1)D* (Tran et al., 2018) combines residual connections with factored space-time 3D convolutions. *Majority Emotion* is a baseline that always predicts Amusement, the majority emotion from the training set. *CLIP* (Radford et al., 2021) trains a joint embedding of images and text, enabling bespoke classifiers. We use Kinetics-400 pretrained versions of STAM and VideoMAE unless otherwise indicated (Kay et al., 2017b). For R(2+1)D, we use pretraining on 65 million weakly-supervised Instagram videos (Ghadiyaram et al., 2019).

**Emotion Prediction.** On the VCE dataset, we train models with the $\ell_1$ loss $\|f(x) - y\|_1$, where $(x, y) \in \mathcal{D}$ is a sample from the training set. We randomly sample clips from each video in the dataset to form a set of clips for a given epoch. We train with minibatches of video clips sampled in this manner for 10 epochs. At test time, we evenly sample multiple clips per video for inference for all models except STAM, which uniformly samples frames instead. We train with a batch size and learning rate of 16 and 0.001 for R(2+1)D and STAM. For CLIP, predictions are zero-shot, and prompted with "The video most strongly evokes", followed by each of the 27 emotions for the text encoder. Additional details are in the Supplementary Material.

We show results on VCE in Tab. 2. Models are compared on the top-3 accuracy metric, which has a random chance level of 11.1% for our dataset. All methods substantially improve upon

|  | Pairwise | | Listwise | |
| --- | --- | --- | --- | --- |
|  | STAM-8 | STAM-16 | STAM-8 | STAM-16 |
| Baseline | 62.8% | 63.4% | 26.9% | 26.7% |
| +VCE | 63.2% | 63.4% | 25.7% | 26.8% |
| +Kinetics | 84.4% | 86.7% | 38.8% | 44.6% |
| +VCE +Kinetics | 84.9% | 86.4% | 38.4% | 43.2% |

Table 3: Wellbeing results on V2V. Pretraining greatly improves performance, although there is still much room for improvement. Random chance for pairwise and listwise accuracy is $50\%$ and $17\%$.

random chance, with the best-performing method being VideoMAE. However, the Majority Emotion predictor attains higher accuracy than CLIP, indicating that zero-shot prediction of emotional responses may be challenging. We find that vision Transformers outperform spatiotemporal convolutions in R(2+1)D, even when the latter is pretrained on 65 million videos. To examine the effect of dataset size on test accuracy, we train STAM-8 with subsets of VCE and plot top-3 accuracy in Figure 8. The $x$-axis denotes thousands of videos in the training set. We find that test performance scales logarithmically with dataset size, and using less than 5,000 videos substantially reduces performance. This highlights the value of the large scale of our datasets.

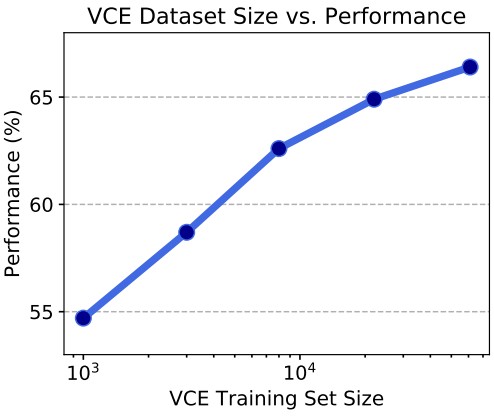

Figure 8: Accuracy on VCE increases logarithmically with the number of training examples. Our large dataset size helps drive high performance.

**Wellbeing Prediction.** On the V2V dataset, we train models to output continuous scores with ranking supervision. This is achieved by letting models output a single, continuous value $f(x)$ on input $x$ and enforcing consistency with all rankings in the training set. For a given ranking $(x_i, x_j, y_{ij})$ in the training set, the training loss is $\text{BCE}\left(\sigma\left(f(x_j) - f(x_i)\right), y_{ij}\right)$, where BCE is the binary cross-entropy. Previous work has used this loss to train utility functions on general scenarios in text (Hendrycks et al., 2020). We focus on STAM models due to their efficiency, evaluating performance on V2V with and without Kinetics pretraining and with different temporal context lengths. The STAM-8 model takes 8 frames as input, and STAM-16 takes 16 frames. We train with batch size of 8 comparisons (16 videos) and learning rate 0.005 for 10 epochs for all models with a single sampling of frames from each video for both training and testing, as described in Sharir et al. (2021).

We show quantitative results on V2V in Tab. 3 and qualitative results in Figure 5. Pairwise accuracy is substantially above random chance, and pretraining on Kinetics results in large improvements, showing that representations for recognizing actions transfer to predicting subjective judgments of relative pleasantness. We experiment with augmenting the training loss with the $\ell_1$ VCE loss scaled by 0.5, but this does not improve performance in all cases. Listwise accuracy is far below pairwise accuracy, and performance on both metrics is far from the ceiling, showing that while models are beginning to gain cognitive empathy and the ability to predict judgments of relative pleasantness, there is still room for improvement.

## 6 Conclusion

We introduced the Video Cognitive Empathy (VCE) and Video to Valence (V2V) datasets for predicting subjective responses to videos. We collected over 60,000 videos and hundreds of thousands of annotations for fine-grained evoked emotions and relative pleasantness. In analyses of our data, we showed that the full distribution of emotional responses on a video is a strong feature for predicting relative pleasantness, suggesting that studying emotions may be important for understanding general preferences over videos. In experiments with state-of-the-art video models, we found that models perform substantially better than chance, although there is still room to improve. As models become better predictors of experienced emotions and factors such as emodiversity, they will become increasingly relevant for monitoring wellbeing.

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
