# How Would The Viewer Feel?
# Estimating Wellbeing From Video Scenarios
# Supplementary Material

**Mantas Mazeika**\*
UIUC

**Eric Tang**\*
UC Berkeley

**Andy Zou**
UC Berkeley

**Steven Basart**
UChicago

**Jun Shern Chan**
UC Berkeley

**Dawn Song**
UC Berkeley

**David Forsyth**
UIUC

**Jacob Steinhardt**
UC Berkeley

**Dan Hendrycks**
UC Berkeley

## A   Additional Related Work

**Value Learning.**   Building machine learning systems that interact with humans and pursue human values may require understanding aspects of human subjective experience. Many argue that values are derived from subjective experience [Hume, 1739, Sidgwick, 1907, de Lazari-Radek and Singer, 2017] and that some of the main components of subjective experience are emotions and valence. Learning representations of values is necessary for creating safe machine learning systems [Hendrycks et al., 2021b] that operate in an open world. In natural language processing, models are trained to assign wellbeing or pleasantness scores to arbitrary text scenarios [Hendrycks et al., 2021a]. Recent work in machine ethics [Anderson and Anderson, 2011] has translated this knowledge into action by using wellbeing scores to steer agents in diverse environments [Hendrycks et al., 2021c]. However, this recent line of work so far exclusively considers text inputs rather than raw visual inputs.

**Emodiversity.**   A large body of work in psychology seeks to understand and quantify the richness and complexity of human emotional life [Barrett, 2009, Lindquist and Barrett, 2008, Carstensen et al., 2000]. An important concept in this area is emodiversity, the variety and relative abundance of emotions experienced by an individual, which has been linked with reduced levels of anxiety and depression [Quoidbach et al., 2014]. Although prior work studies emodiversity in self-reports of emotion without stimuli, we hypothesize that the emodiversity of visual stimuli may be an important concept to quantify and understand. Thus, we examine how our new datasets could enable measuring the emodiversity of in-the-wild videos on a large scale.

## B   Data Collection

We collect videos for VCE and V2V from manually selected online sources on Reddit and Instagram with high potential to evoke emotions. The videos were scraped by undergraduate and graduate student authors. Upon receiving IRB approval, annotations of subjective experience are gathered from 400 annotators on Amazon Mechanical Turk who passed a qualification process. For example, one of the qualification questions for VCE was "A person sees someone stub their toe. An emotion they may experience is (A) Awe, (B) Excitement, (C) Calmness, (D) Empathetic Pain" (correct answer: D). The qualification process ensures that annotators are paying attention and understand the questions. The resulting pool of annotators were primarily from the US, who were compensated 35 cents for each task. Upon passing the qualification process, annotators were given the following instructions.

---

\*Equal Contribution.

36th Conference on Neural Information Processing Systems (NeurIPS 2022) Track on Datasets and Benchmarks.

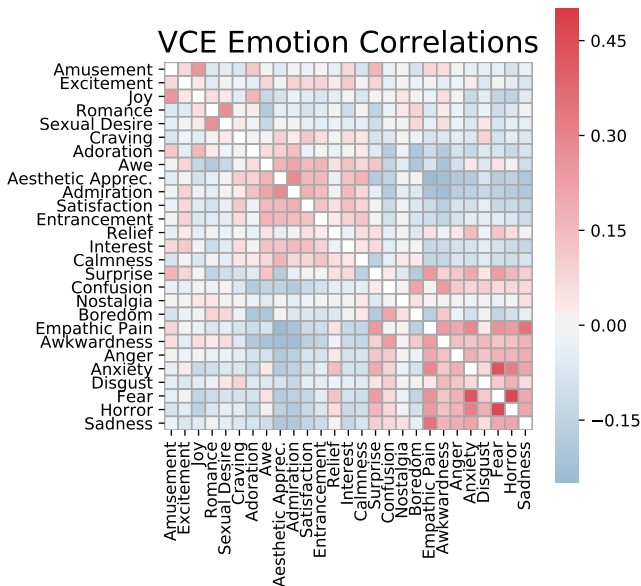

Figure 1: Emotional responses in VCE are correlated in reasonable ways. For example, awe, aesthetic appreciation, admiration, satisfaction, and entrancement are all weakly correlated, reflecting the fact that these emotions can overlap on a given video as different annotators may have different subjective experiences of the video. In this correlation matrix, we mask out the diagonal.

## B.1 VCE Instructions

In this study, you will see 15 videos. Alongside each video will be 27 emotions. Select at least one or more than one emotions that capture how each video makes you feel. You can select multiple emotions for a video.

A different video will appear as you go to each page of this survey. Each video will play once automatically on load- you can replay the video by clicking the play button in the bottom left of the video. Please watch each video in its entirety at least once before responding to it.
If the videos fail to appear, do not submit this HIT.
Use the buttons below each video to choose emotions that describe how it makes you feel. There are 27 buttons. Choose one or more than one emotions as needed to describe your emotional response(s). You can choose several emotions for each video.
Once you select an emotion for a video a slider will appear for that emotion, with the default value set to 10. Adjust the slider on the scale from 1 - 10 based on how strongly the video evokes the corresponding emotion, with 10 meaning the video strongly evokes that emotion, and 1 meaning the video only slightly evokes that emotion. Do adjust sliders appropriately.
Since most internet videos are somewhat amusing, if you pick "Amusement" as an emotion for a video, you must also select another emotion in addition to it.
If you believe you can't understand the emotional response to a video without its audio, do not select any emotions, and check the box saying "Invalid video - relies on audio." You can still submit if the video relies on audio; if you do not see any video, do not submit and return the HIT.
If you choose randomly you will be banned and rejected. We actively look at responses to find random responses. It is very obvious when submissions are random.
After you have selected at least one or more emotions for each video in this HIT, click the continue button until there are no more videos to be rated.

Here are the 27 emotions and their rough meaning:

1. **Admiration** – a feeling of respect for and approval of somebody/something

2. **Adoration** – a feeling of great love

3. **Aesthetic Appreciation** – pleasure that you have when you recognize and enjoy the good qualities of how something looks

4. **Amusement** – the feeling that you have when you enjoy something that is entertaining or funny

5. **Anger** – the strong feeling that you have when something has happened that you think is bad and unfair

6. **Anxiety** – the state of feeling nervous or worried that something bad is going to happen

7. **Awe** – feelings of respect and slight fear; feelings of being very impressed by something/somebody

8. **Awkwardness** – feelings or signs of shame or difficulty

9. **Boredom** – the state of feeling bored; the fact of being very boring

10. **Calmness** – the quality of not being excited, nervous or upset

11. **Confusion** – a state of not being certain about what is happening, what you should do, what something means, etc.

12. **Craving** – a strong desire for something

13. **Disgust** – a strong feeling of dislike for somebody/something that you feel is unacceptable, or for something that has an unpleasant looks, smell, etc.

14. **Empathic Pain** – to feel pain by understanding another person's feelings and experiences

15. **Entrancement** – enchanting and a feeling of delight

16. **Excitement** – the state of feeling or showing happiness and enthusiasm

17. **Fear** – the bad feeling that you have when you are in danger or when a particular thing frightens you

18. **Horror** – an overwhelming and painful feeling caused by something frightfully shocking, terrifying, or revolting

19. **Interest** – the feeling that you have when you want to know or learn more about somebody/something

20. **Joy** – a feeling of great happiness

21. **Nostalgia** – a sad feeling mixed with pleasure when you think of happy times in the past

22. **Relief** – the feeling of happiness that you have when something unpleasant stops or does not happen

23. **Romance** – love or the feeling of being in love

24. **Sadness** – the feeling of being sad

25. **Satisfaction** – the good feeling that you have when something that you wanted to happen does happen

26. **Sexual Desire** – a desire for sexual intimacy

27. **Surprise** – an event, a piece of news, etc. that is unexpected or that happens suddenly

## B.2  V2V Instructions

In this study, you will see 15 pairs of videos. Alongside each pair will be 4 options for you to pick from in order to rate the relative pleasantness of the videos, going from strongly preferring the first video displayed, to slightly preferring the first video, then to slightly preferring the second video, and finally to strongly preferring the second video.

We will also give you the option to abstain from rating a pair if you feel that it is unclear which you and other MTurkers would prefer to watch. However, you may only use this option once per HIT.

For the following video pairs: which video do you think other MTurkers would think is the most pleasant (and least unpleasant)? If uncertain, which do you think is most pleasant? Watch the video in its entirety and evaluate the video overall/holistically, not necessarily the feeling you had at the middle of the video. Which was most pleasant to watch?

Something that may help is imagining that you were there if appropriate for the video (would not be for highly edited/cartoon videos). How would you feel if you were there?

You may use 1 skip for a pair of clips you are very uncertain about per HIT

A different pair of videos will appear as you go to each page of this survey. Each video will play on loop.
Use the radio buttons below the videos to select which video you believe other MTurkers would find more pleasant. If the videos fail to appear, do not submit this HIT.
If you choose randomly you will be banned and all of your HITs rejected. We actively look at responses to find random responses.
After you have ranked each pair of videos for this HIT, click the continue button to finish.

Please rewatch videos that you think you'd do a better job assessing them if you watched them again.
If both videos are unpleasant, which is least unpleasant?
We are not asking what is most weird, entrancing, surprising, but instead what is overall most pleasant.
If uncertain, the following may also help: Imagine you were in the video next to the camera person. How are you feeling? (For fake scenarios, say how pleasant it is to observe.)

### B.3   V2V Dataset Construction

Pairs for the V2V dataset were selected primarily based on labels from the VCE dataset. The main strategy used for sampling was to consider the $\ell_1$ distance between both the ground truth labels as well as model predictions on pairs of videos, averaged over the highest performing models that we ran experiments on (an ensemble of TimeSformer and STAM). Pairs of videos that had a large $\ell_1$ distance from one another based on the VCE dataset, but that contained model predictions that had a relatively smaller $\ell_1$ distance make up a large portion of the final dataset. In addition to this strategy, we also experimented with sampling pairs of videos randomly, as well as sampling pairs solely based on having similar ground truth labels or similar predictions from models trained on VCE, in order to encourage interesting comparisons between videos.

### B.4   Data Sources

We collect videos with the following Instagram hash tags: adorable, adorablevideos, aestheticvideos, artvideos, beautifulvideos, bunniesofinstagram, calmingvideos, caughtoncamera, closecall, cookingvideos, coolvideo, couplevideos, creepyvideo, cutemoments, drawingvideo, epicscene, epicvideo, failvideo, funnyvideos, hairvideos, happyvideo, horrorvideo, illusionvideo, interestingvideo, magicvideo, moodyvideo, proposalvideo, sadvideos, satisfyingvideos, sciencevideos, sportsvideo, trendingvideo, videography, videooftheday, videostar, viralvideos, weirdvideos, workoutvideos.

We collect videos from the following subreddits: animalsbeingderps, animalsbeingjerks, art, aww, BetterEveryLoop, calm, CatastrophicFailure, catvideos, Damnthatsinteresting, creepyvideos, EAF, fastworkers, FoodVideos, funny, funnygifs, funnyvideos, gifs, HadToHurt, HorriblyDepressing, IdiotsInCars, instant_regret, InterestingVideoClips, JusticeServed, KidsAreFuckingStupid, MadeMeCry, maybemaybemaybe, mildlyinfuriating, NatureGifs, NatureIsFuckingLit, nextfuckinglevel, nonononoyes, oddlysatisfying, opticalillusions, PublicFreakout, rage, RelaxingGifs, sad, sadcringe, trippyvideos, unexpected, WatchPeopleDieInside, Whatcouldgowrong, woahdude, WTF, yesyesyesyesno.

The annotated emotions in VCE correlate with the data source in reasonable ways. For instance, the most common annotated emotions across videos from the subreddits "funny" and "fastworkers" are amusement and admiration, respectively. However, the per-video annotations have significant variance across annotators, reflecting the breadth of human emotional responses.

## C   Experiment Details

For the emotion prediction task on the VCE dataset, we primarily use Vision Transformer based models pretrained on Kinetics-400. We use standard data transformations, resizing any input image to 256x256, then taking a center crop for a final input shape of 224x224. We use Nesterov accelerated gradient descent with momentum $0.9$, and a cosine annealing learning rate, with learning rate initially

# Preferences ≠ Choices ≠ Wellbeing

what people want
intentions, motives

what people do
engagement, behavior

what happens as a result
welfare of user, value to user

Figure 2: Choices are easy to measure and reveal some information about preferences. However, they do not perfectly reveal preferences. Moreover, satisfying preferences does not always lead to increasing wellbeing. Thus, grounding economics in revealed preferences may lead to a stagnation in wellbeing past a certain point. Consequently, we need better tools for directly measuring wellbeing. AI could provide a way to efficiently measure true wellbeing at scale.

set to $1 \times 10^{-2}$. For inference, we use 10 clips evenly spaced over the video for all models except for STAM, for which we use 1 set of frames evenly spaced across the video.

## D  Legal Sourcing and Intended Usage

The videos in VCE and V2V are publicly available and downloaded from Reddit and Instagram. Some videos may be under copyright. Hence, we follow Fair Use §107: "the fair use of a copyrighted work, including such use by ... scholarship, or research, is not an infringement of copyright", where fair use is determined by "the purpose and character of the use, including whether such use is of a commercial nature or is for nonprofit educational purposes", "the amount and substantiality of the portion used in relation to the copyrighted work as a whole", and "the effect of the use upon the potential market for or value of the copyrighted work." Hence, the VCE and V2V datasets are noncommercial and should only be used for the purpose of academic research.

**Additional Usage Considerations and Broader Impacts.** The VCE and V2V datasets are designed to give a high-level understanding of how well current video models can predict subjective responses to videos. In particular, we do not design the datasets to enable conditioning on cultural background or personality traits, which strongly influence emotional responses and preferences [Lim, 2016, Hoerger and Quirk, 2010]. Hence, our annotations should not be taken to represent accurate emotional responses across a broad range of cultures or on an individual level, and we discourage their use in deployment contexts. We support work on large-scale data collection that considers differences in emotional responses across cultures and individuals, and we think this is an interesting direction for future research.

We do not support the potential usage of emotion prediction datasets for socially harmful applications, such as addictive content recommendation or persuasion. In particular, we recognize the possibility for emotional response prediction to render persuasive media more effective and worsen its negative effects. However, we hope that the net impact of datasets measuring the ability of models to predict emotional responses will be positive. This is because it is important to know whether pretrained video models have this capability in the first place, and because there are numerous positive applications for this technology, such as counteracting highly addictive engagement-based content recommendation. Nevertheless, it is important to monitor the usage of this technology as it develops and consider possible regulations to restrict negative use cases.

### D.1  Author Statement and License.

We bear all responsibility in case of violation of rights. Some of the videos in the VCE and V2V datasets may be under copyright, so we do not provide an official license and rely on Fair Use §107. Our code is open sourced under the MIT license. Our annotations are available under a CC BY-SA 4.0 license.

## E  X-Risk Sheet

We provide an analysis of our paper's contribution to reducing existential risk from future AI systems following the framework suggested by [Hendrycks and Mazeika, 2022]. Individual question responses

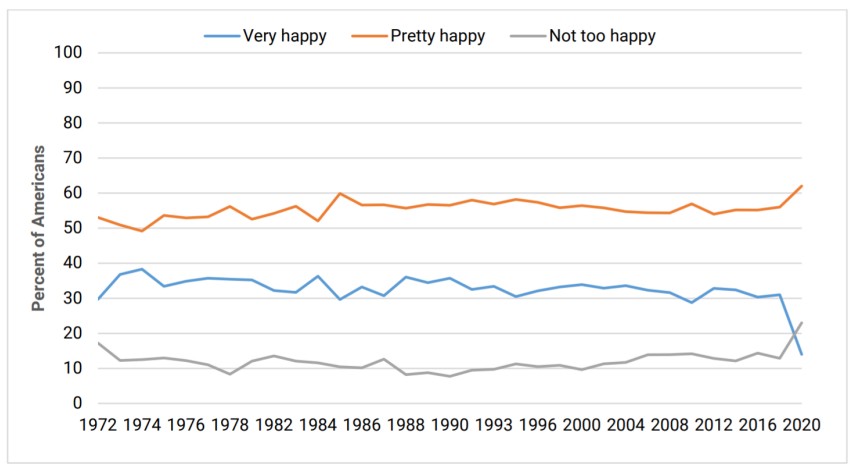

Figure 3: Happiness in the US has not increased since 1970 and has recently decreased due to the pandemic (figure source: [NORC, 2020]). In the same period, real GDP per capita more than doubled [FRED]. This indicates that maximizing wealth may be a poor proxy for maximizing wellbeing.

do not decisively imply relevance or irrelevance to existential risk reduction. Do not check a box if it is not applicable.

### E.1 Long-Term Impact on Advanced AI Systems

In this section, please analyze how this work shapes the process that will lead to advanced AI systems and how it steers the process in a safer direction.

1. **Overview.** How is this work intended to reduce existential risks from advanced AI systems?
   **Answer:** This work is intended to reduce risks from proxy misspecification in advanced AI systems. In this work, we build datasets for developing AI systems that can predict human emotional responses and pleasantness rankings over a wide range of in-the-wild video scenarios. A central goal of AI safety is to enable the alignment of future AI systems with human values. Emotions are an important aspect of human values, because values are derived from subjective experience, and a large part of human subjective experience involves emotions. Additionally, emotions can be thought of as evaluations of events in relation to goals and thus are directly useful for understanding the goals of individuals. Finally, pleasure is the foremost intrinsic good, so building strong predictive models of pleasantness for arbitrary scenarios is an important component of modeling human values. Thus, AI systems that can accurately predict and understand the values and subjective experiences of humans would enable better proxy objectives for a wide range of applications.

2. **Direct Effects.** If this work directly reduces existential risks, what are the main hazards, vulnerabilities, or failure modes that it directly affects?
   **Answer:** This work directly reduces AI system risks from proxy misspecification.

3. **Diffuse Effects.** If this work reduces existential risks indirectly or diffusely, what are the main contributing factors that it affects?
   **Answer:** By enabling the measurement of how well AI systems can predict human emotional responses, this work could encourage the adoption of standards or regulations regarding the ability of strong AIs to understand human values. By enabling iterative improvement on this task, we hope to improve safety culture by making it easier for other researchers to measure performance and improve on the task

4. **What's at Stake?** What is a future scenario in which this research direction could prevent the sudden, large-scale loss of life? If not applicable, what is a future scenario in which this research direction be highly beneficial?
   **Answer:** If future strong AI systems are used to design national policies in a democracy, it is imperative that they are in touch with the entire constituency and understand what they value. If a possible negative side effect of a policy could substantially harm a group of individuals, the AI

should be aware of this and factor it into decision-making. Otherwise it could end up proposing suboptimal policies that needlessly harm large numbers of people, up to and including loss of life.

5. **Result Fragility.** Do the findings rest on strong theoretical assumptions; are they not demonstrated using leading-edge tasks or models; or are the findings highly sensitive to hyperparameters? ☐

6. **Problem Difficulty.** Is it implausible that any practical system could ever markedly outperform humans at this task? ☐

7. **Human Unreliability.** Does this approach strongly depend on handcrafted features, expert supervision, or human reliability? ☒

8. **Competitive Pressures.** Does work towards this approach strongly trade off against raw intelligence, other general capabilities, or economic utility? ☐

## E.2 Safety-Capabilities Balance

In this section, please analyze how this work relates to general capabilities and how it affects the balance between safety and hazards from general capabilities.

9. **Overview.** How does this improve safety more than it improves general capabilities?
**Answer:** Designing systems to better predict human emotional responses is unlikely to improve general capabilities. While it is possible that predicting emotional responses in videos is a uniquely challenging task that could lead to the development of general improvements in video understanding, it is probably possible to improve performance on the task without improving general video understanding capabilities, e.g., by collecting more data of human emotional responses or incorporating theoretical frameworks as inductive biases. By providing datasets for measuring performance on this task, we hope to encourage the latter kind of work. It is also valuable to track how well general improvements in video understanding transfer to improving predictions of emotional responses, so our work could still improve the safety-capabilities balance even if marginal gains on the task are challenging.

10. **Red Teaming.** What is a way in which this hastens general capabilities or the onset of x-risks?
**Answer:** Predicting emotional responses in videos may be a uniquely challenging task that could lead to the development of general improvements in video understanding.

11. **General Tasks.** Does this work advance progress on tasks that have been previously considered the subject of usual capabilities research? ☐

12. **General Goals.** Does this improve or facilitate research towards general prediction, classification, state estimation, efficiency, scalability, generation, data compression, executing clear instructions, helpfulness, informativeness, reasoning, planning, researching, optimization, (self-)supervised learning, sequential decision making, recursive self-improvement, open-ended goals, models accessing the Internet, or similar capabilities? ☐

13. **Correlation With General Aptitude.** Is the analyzed capability known to be highly predicted by general cognitive ability or educational attainment? ☐

14. **Safety via Capabilities.** Does this advance safety along with, or as a consequence of, advancing other capabilities or the study of AI? ☐

## E.3 Elaborations and Other Considerations

15. **Other.** What clarifications or uncertainties about this work and x-risk are worth mentioning?
**Answer:** Regarding Q6, one can reach superhuman performance at predicting emotional responses or valence (assuming that humans are limited to typical interactions). Heart rate tracking enables accurate measurement of whether an individual is nervous or surprised, and EEG signals contain reliable information about emotional responses.

Regarding Q7, predicting human emotional responses necessitates a dataset of emotional responses obtained from humans. Thus, unreliability of self-report is an issue that one must contend with. However, this could be mitigated by using less subjective measuring devices, including heart rate monitors and EEGs.

Regarding Q11 and Q12, while the VCE and V2V datasets could indirectly lead to general capabilities advancements, we view this as highly unlikely.

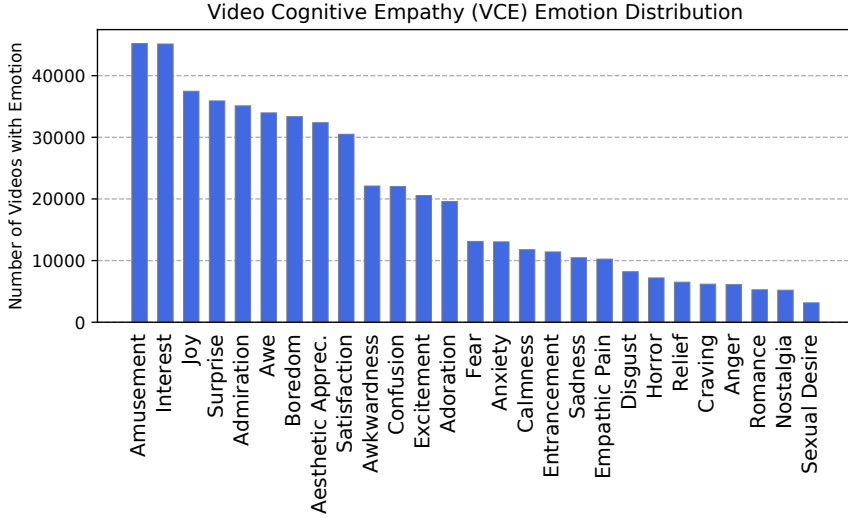

Figure 4: Statistics of the Video Cognitive Empathy dataset. Emotional responses span a wide range of categories, with a greater focus on emotions with positive valence.

Regarding Q13, most humans are quite good at identifying or predicting the emotional responses of other people, so the ability is not strongly correlated with general cognitive ability or educational attainment.

Finally, we would like to discuss several points regarding the broader importance of building AI systems that can measure wellbeing at scale.

(a) The concept of preference is central in economics, with methods such as revealed preference theory and the often held assumption that maximizing wealth (and thereby satisfaction of revealed preferences) is a good proxy for maximizing the actual welfare of participants in an economy. However, there is much disagreement around these assumptions on how to ground economics, including from high-profile voices in economics such as Nobel Prize laureate Amartya Sen [Anderson, 2001]. Empirically, wealth maximization is not helping happiness in the first-world. Consider Figure 3; past a certain point, ability to satisfy one's preferences via economic power does not translate into increases in true wellbeing. In other words, maximizing wealth is not the same as maximizing wellbeing, and maximizing wealth on an individual level may have at best exponentially diminishing returns to wellbeing [Kahneman and Deaton, 2010].

(b) Revealed preference theory is based on the idealistic assumption that people flawlessly take actions in order to rationally satisfy their preferences. The rise of behavioral economics was a response to the empirical observation that in fact, people do not [Kahneman et al., 1982]. In his 2002 Nobel Prize in Economics lecture, Kahneman addressed these failures of classical economic theory, saying, "A theory of choice that completely ignores feelings such as the pain of losses and the regret of mistakes is not only descriptively unrealistic, it also leads to prescriptions that do not maximize the utility of outcomes as they are actually experienced—that is, utility as Bentham conceived it." [de Lazari-Radek and Singer, 2014].

(c) People engage in activities that they choose that can harm welleing–consider addiction. Additionally, people may possess preferences that do not align with what would actually maximize their wellbeing–consider how an individual following their preferences to browse social media all day may not actually be maximizing their short-term or long-term wellbeing [Kross et al., 2013]. Thus, choices do not perfectly reveal preferences, and preferences do not always correspond to wellbeing, as illustrated in Figure 2.

(d) Economics should be founded on wellbeing, not preferences. Thus, better tools for measuring wellbeing would be highly valuable. In this work, we demonstrate that AI is a promising tool for measuring important aspects of wellbeing.