# OpenReview forum: "How Would The Viewer Feel? Estimating Wellbeing From Video Scenarios"
_NeurIPS.cc/2022/Track/Datasets_and_Benchmarks — NeurIPS 2022 Datasets and Benchmarks _

### Official Review · Reviewer_c8mi · 2022-07-25
**Great dataset, solid paper**

**Rating:** 9
**Confidence:** 4
**Clarity:** The manuscript is written very clearly.

**Strengths:**

The utility of the datasets is very clear, both for direct application as well as more foundational AI research. The datasets are large and constitute a major contribution.

**Weaknesses:**

I could not identify any signficant weaknesses in this work.

**Additional Feedback:**

N/A

**Correctness:**

All claims made seem correct.


**Documentation:**

The documentation seems good.

**Ethics:**

The paper aims to provide a depiction of emotional responses to video content. As such, there are potential concerns about how well (or not) the global diversity of societal backgrounds and worldviews are represented by the cohort of Amazon MTurk annotators (which might correlate with emotional responses to some content). Although this concern is somewhat lessened by the fact that the dataset contains the full distribution of responses, the paper should briefly mention any potential limitations associated with this concern.

The inteded usage section in the supplementary material could expand a bit more on how such a dataset might be used or misused in practical applications (i.e. a social media site might try to use the dataset to train algorithms ranking user-generated content)

**Relation To Prior Work:**

The relation to previous work is established well through in the related work section.

**Summary And Contributions:**

The authors introduce two datasets, Video Cognitive Empathy (VCE) and Video to Valence (V2V). The datasets contain over 60,000 short videos that were annotated with emotional content and general emotional valence by crowdworkers. The authors train machine learning models on the datasets and demonstrate that some models reach good performance, but that there is also room left for further improvement.

---

> ### Author Response · Authors · 2022-08-20
> **Response to Reviewer c8mi**
>
> Thank you for your careful analysis of our work. We hope the following response addresses your concerns.
>
> **Expanded Intended Usage Section.**
>
> The intended usage section in the original submission was unfortunately cut off due to a hard-to-spot LaTeX error. We have restored the remaining text in the updated paper, which includes a disclaimer that “our annotations should not be taken to represent accurate emotional responses across a broad range of cultures or on an individual level, and we discourage their use in deployment contexts”. Thanks to your suggestion, we have also added text clarifying that we do not support the potential use of emotion prediction datasets for socially harmful applications, such as addictive content recommendation or persuasion. However, we do believe that careful usage of deployment-quality datasets for emotion prediction could yield socially beneficial applications, such as improved content search and screening videos for distressing moments. Thank you for bringing this to our attention and for your suggestions.

---

> > ### Comment · Reviewer_c8mi · 2022-08-24
> > **Thanks**
> >
> > Thanks for the correction.
> > I have no further comments, my initial recommendation for acceptance remains unchanged.

---

### Official Review · Reviewer_58RH · 2022-07-28

**Rating:** 8
**Confidence:** 4
**Correctness:** The experiments seem correct.
**Clarity:** The paper is well written.

**Strengths:**

- The paper highlights the motivation and idea in very efficient manner.
- This paper introduces two large dataset to understand the human emotions after watching a video.

**Weaknesses:**

- The background of 400 annotators are not mentioned in the paper, if the majority of people are from a single locality/country then the emotions annotated by them can be biased. As population from one diversity can feel completely different from populatio from different diversity.
- If the videos contains mixed emotions like first sad then happy, such type of videos will have very subjective label depending on the person.

**Additional Feedback:**

- Emotions based on different diversity people can be collected.

**Documentation:**

The dataset description and availability is complete.

**Ethics:**

No ethical issues.

**Relation To Prior Work:**

Properly mentioned in the paper.

**Summary And Contributions:**

For understanding how viewers feel while watching videos, the authors introduce two large-scale datasets for predicting emotional state and wellbeing of viewers directly from videos. The dataset contains 60,000 videos with human annotations for 27 emotion categories.

---

> ### Author Response · Authors · 2022-08-20
> **Response to Reviewer 58RH**
>
> Thank you for your careful analysis of our work. We hope the following response addresses your concerns.
>
> **Intended Usage of the Datasets.**
>
> We agree that the background of annotators is an important factor that can bias responses. The annotators for VCE and V2V were not drawn from a globally representative population, so we do not endorse the usage of VCE and V2V in deployment contexts. Rather, the datasets are meant to be used in an academic context and are designed to give a high-level understanding of how well current video models can predict subjective responses to video. We discuss intended usage for the datasets in more detail in the Intended Usage section of the appendix.
>
> **Subjective Nature of Labeling Emotions.**
>
> We take several measures to acknowledge and account for the subjective nature of labeling the emotional content of videos. For videos with mixed emotions, you are correct that different annotators may pick different emotions. Our annotations for VCE are distributions of responses across multiple annotators, which enables modeling this variation. For V2V, we only use video pairs where there is high inter-annotator agreement.

---

### Official Review · Reviewer_JE1G · 2022-07-30
**Largescale Multilabel Datasets for Predicting Emotional Responses and Degree of Pleasantness**

**Rating:** 7
**Confidence:** 4
**Correctness:** Authors seem to have quite diligently…
**Clarity:** Yes, the paper is well written.

**Strengths:**

Although, not altogether novel (since there have similar smallscale datasets), these datasets are largest of their kinds, which would allow end-to-end representation learning.
In my opinion, these building such dataset would require significant efforts, which adds to the value/contribution of these datasets. Authors have diligently designed dataset collection/annotation protocols. Although, please see Weaknesses section.

**Weaknesses:**

Annotators were allowed to use audio signals in order to annotate, so the labels might not be guaranteed to have been grounded in video signal alone. Although, authors did ask annotators to take notes of samples for which they relied heavily on the audio signal. I think for annotators making such choices might be somewhat ambiguous. Such ambiguity might be ultimately reflected in annotations as well. Can authors please lend their views in this regard?

Also, authors remove the audio signal from the final dataset. Why not provide it to allow multimodal analysis if desired? In my opinion, removing audio signals just hurts the overall utility of the dataset. Unless, the authors want to get another publication (saying this with all due respect) by enhancing this dataset with audio included version of the dataset. I think authors should provide the accompanying audios.

**Additional Feedback:**

N/A

**Documentation:**

Provided. Although not checked in detail.

**Ethics:**

Systems trained on such datasets could be used to analyze viewers' emotional state, which can raise privacy issues.

**Relation To Prior Work:**

Sufficiently discussed prior work.

**Summary And Contributions:**

This paper proposes two datasets: VCE (samples containing emotion label distributions) and V2V containing pairwise comparative labels in terms of degree of pleasantness. These datasets can be used to predict emotional responses, and wellbeing of viewers. Authors train standard CNNs on these datasets, and observed that while these networks perform well, they do leave a significant performance gap to be fulfilled.

---

> ### Author Response · Authors · 2022-08-20
> **Response to Reviewer JE1G**
>
> Thank you for your careful analysis of our work. We hope the following response addresses your concerns.
>
>
> **Availability of Audio to Annotators.**
>
> Annotators do not have access to audio when annotating, but we do ask annotators to flag whether a video would require audio in order to elicit an emotional response (e.g., for videos that are mainly text on a screen describing a story, presumably with music in the background). Thus, annotators and models see the same data, and there is no asymmetry of information. We have updated the paper to clarify this.
>
> **Reasons for Video-Only.**
>
> We are primarily interested in understanding how emotions depend on the semantic content of videos and less on how engineered cues such as background music can evoke desired emotions. We agree that a careful multimodal analysis that disentangles these factors would be interesting, but this would require special considerations during scraping and annotation, so we chose to focus our resources on the still interesting case of video-only predictions. Additionally, we did not save audio information during the initial scraping process, and reacquiring this information would be challenging.

---

### Review · Ethics_Reviewer_cvrq · 2022-08-21

**Recommendation:** 2

**Ethics Documentation:**

See the ethics review.

**Ethics Review:**

The paper raises a number of ethical concerns (as mentioned already by some of the reviewers) that should be addressed by the authors (or at least for which there should be a plan to address them for the camera ready version) before the paper can be considered for acceptance:
- Models trained on the dataset could be used to predict someone's emotional state. This could have a number of negative societal impacts (eg worsening the effects of manipulative media) which should be mentioned in the paper, including a number of potential mitigations. The authors provide some information on the intended usage but, in this case, I think that this should be elaborated on and should be also (or instead) addressed in the main paper.
- The dataset raises a number of diversity and bias concerns. What is the diversity of people shown in the video? Most of the videos come from Reddit which is likely to predominantly American. And what is the diversity of the cohort of MTurk reviewers? The paper should have some mention of this issue and either a characterisation of the bias or an explanation as to how this should be characterised or accounted for.
- It is unclear to me if the people in the videos have consented to their videos being used. Eg there a videos taken from subreddits like "KidsAreFuckingStupid". Can the authors please comment on whether consent is necessary and why or why not?

---

> ### Author Response · Authors · 2022-08-22
> **Response to Ethics Reviewer cvrq**
>
> Thank you for your careful analysis of our work. We hope the following response addresses your concerns.
>
> **Added Broader Impacts Section.**
>
> We have added a discussion of broader impacts to the appendix, which states:
>
> “We do not support the potential usage of emotion prediction datasets for socially harmful applications, such as addictive content recommendation or persuasion. In particular, we recognize the possibility for emotional response prediction to render persuasive media more effective and worsen its negative effects. However, we hope that the net impact of datasets measuring the ability of models to predict emotional responses will be positive. This is because it is important to know whether pretrained video models have this capability in the first place, and because there are numerous positive applications for this technology, such as counteracting highly addictive engagement-based content recommendation. Nevertheless, it is important to monitor the usage of this technology as it develops and consider possible regulations to restrict negative use cases.“
>
> We hope this change clarifies the potential positive/negative impacts of the research avenue of emotional response comprehension and prediction. On net, we think this research avenue could have a positive impact; we outline several reasons for this in the main paper and the new broader impacts section; the main benefits come from reducing the harmful impacts of recommendation and optimization of content based on shallow engagement metrics. But we also recognize that there are negative use cases, and we discuss several possible mitigation strategies (monitoring and regulation). Thank you for your suggestion
>
>
> **Expanded Discussion of Intended Usage and Dataset Bias.**
>
> We have elaborated on our discussion of intended usage in the appendix and included a substantial portion of this discussion in the main paper. In particular, we clarify that the datasets are not representative of a wide range of cultural backgrounds and should not be used in deployment contexts. Rather, they are meant to give researchers a high-level understanding of how well current video models can predict human emotional responses. The text that we added to the main paper is reproduced below.
>
> “Note that the MTurk annotators and individuals depicted in the videos may not form a representative sample of diverse cultural backgrounds. Hence, our annotations should not be taken to represent accurate emotional responses across a broad range of cultures or on an individual level, and we discourage their use in deployment contexts. The VCE and V2V datasets are designed to give a high-level understanding of how well current video models can predict subjective responses to videos. We support work on large-scale data collection that considers differences in emotional responses across cultures and individuals, and we think this is an interesting direction for future research.“
>
> We hope this change clarifies the limitations of our dataset and its intended usage. Thank you for your suggestion
>
>
> **Consent for In-The-Wild Videos.**
>
> We agree that a central consideration when building new datasets is legal and ethical sourcing of data. Ideally, all individuals depicted in our datasets would give fully informed consent. However, one of the challenges with in-the-wild data is that it is extremely hard to put this infrastructure in place. An advantage of in-the-wild data is that one doesn’t have to deal with biases created by actors or constructed scenarios, and one can obtain much greater diversity. In our case, we felt that the advantages outweighed the disadvantages because of the long tail of scenarios that can elicit emotional responses (e.g., there are only a few ways to ride a bike, but there are many ways to feel joy).
>
> Historically, we have published several papers at NeurIPS and satisfied their ethics reviews, and each time we were legally compliant. In short, all the videos in VCE may be under copyright by the original owners. We respect these rights by following Fair Use §107: "the fair use of a copyrighted work, including such use by ... scholarship, or research, is not an infringement of copyright", where fair use is determined by "the purpose and character of the use, including whether such use is of a commercial nature or is for nonprofit educational purposes", "the amount and substantiality of the portion used in relation to the copyrighted work as a whole", and "the effect of the use upon the potential market for or value of the copyrighted work." In the paper, we specify that VCE and V2V are noncommercial datasets intended purely for academic use. We also explicitly state the license (CC for the MTurk annotations) and intended use cases. Thank you for raising these important points.

---

> > ### Comment · Reviewer_9xkh · 2022-08-23
> > **Thank you for the elaborate response**
> >
> > Thank you for the elaborate response!
> >
> > My concerns have been largely addressed. However, I feel that the potential negative societal impacts and concerns around lack of representation are so important for this work that I would strongly encourage the authors to move this section to the main paper, if the paper gets accepted. The camera ready version will allow for an additional page.
> >
> > Additionally, I wonder if it is not possible to give some statistics around the representation. Perhaps the authors have more information about the pool of MTurk participants? Were there, for example, any filters applied to select participants?
> >
> > Finally, and apologies for not raising this point earlier, I noticed that in the checklist, the authors do state the compensation in terms of "cents" per task. Could do the authors give some indication of the average hourly wage that participants earned?

---

> > > ### Author Response · Authors · 2022-08-26
> > > **Response to Reviewer 9xkh**
> > >
> > > Thank you for your additional comments. We hope the following response addresses your remaining concerns.
> > >
> > > **Added Discussion of Bias to Main Paper.**
> > >
> > > We have moved the discussion of bias in the dataset to the main paper. This uses up the additional 10th page that would be allowed if the paper is accepted. Thank you for your suggestion.
> > >
> > > **Additional Information on MTurk annotation process.**
> > >
> > > For VCE, we used a set of unambiguous qualification questions to filter out workers who were not paying attention or who were unable to understand the questions. For example, one of the qualification questions was “A person sees someone stub their toe. An emotion they may experience is (A) Awe, (B) Excitement, (C) Calmness, (D) Empathetic Pain” (correct answer: D). For V2V, we used unambiguous annotations for the relative pleasantness of a small set of videos as qualifying questions. The MTurk workers who passed these qualifications ended up being primarily from the United States. We did not collect demographic information from the annotators, but there is some evidence that the demographic distribution of MTurkers from the US is quite diverse: https://www.cloudresearch.com/resources/blog/who-uses-amazon-mturk-2020-demographics/
> > >
> > > We compensated workers 35 cents per task. Each task entailed annotating several videos or video pairs, and we estimate that workers could have earned 4 to 5 dollars/hour performing these tasks. For comparison, the median hourly MTurk wage in 2017 was approximately 3.18 dollars/hour: https://arxiv.org/pdf/1712.05796.pdf
> > >
> > > We have updated the paper and datasheet to include the above information.

---

### Comment · Reviewer_c8mi · 2022-07-15
**Dataset availability**

The manuscript states "Our datasets and experiment code can be found at github.com/hendrycks/emodiversity."
Unfortunately, this URL results in a 404 error. Please fix and/or comment!

---

> ### Author Response · Authors · 2022-07-15
> **Dataset repository is now public**
>
> Hello,
>
> We have now changed the repository to a public repository, so you should be able to access the dataset. Thank you for bringing this to our attention.

---

### Meta-Review · Area_Chair_5SYF · 2022-09-09

**Recommendation:** Accept
**Confidence:** 4

**Metareview:**

Overall, the paper is capturing a really timely and increasingly important topic. Although the authors mention that they want to make their dataset particularly useful for academics/future research, I believe, that the topic is of great interest for companies, making the raised ethical considerations even more important. A critical reflection upon possibly ‘unintended uses’ can’t be emphasised enough, as well as possible biases in the data. However, I appreciated that the authors have done a fantastic job in their response and strengthening their article. Moreover, all reviewers recognise the utility and relevance of the proposed dataset and the authors did provide satisfactory responses to all raised concerns/comments. I believe this paper will stimulate some interesting and hopefully critical reflections on how to use/or not use the dataset and future improvements to overcome dataset biases.

---

### Decision · Program_Chairs · 2022-09-16

Accept